# Estimating weekly excess mortality at sub-national level in Italy during the COVID-19 pandemic

Marta Blangiardo[1]*, Michela Cameletti[2], Monica Pirani[1], Gianni Corsetti[3], Marco Battaglini[3], Gianluca Baio[4]

**1** MRC Centre for Environment and Health, Department of Epidemiology and Biostatistics, Imperial College London, London, United Kingdom, **2** Department of Management, Economics and Quantitative Methods, University of Bergamo, Bergamo, Italy, **3** Istat, Directorate for Social Statistics and Population Census, Rome, Italy, **4** Department of Statistical Sciences, University College London, London, United Kingdom

* m.blangiardo@imperial.ac.uk

## Abstract

In this study we present the first comprehensive analysis of the spatio-temporal differences in excess mortality during the COVID-19 pandemic in Italy. We used a population-based design on all-cause mortality data, for the 7,904 Italian municipalities. We estimated sex-specific weekly mortality rates for each municipality, based on the first four months of 2016–2019, while adjusting for age, localised temporal trends and the effect of temperature. Then, we predicted all-cause weekly deaths and mortality rates at municipality level for the same period in 2020, based on the modelled spatio-temporal trends. Lombardia showed higher mortality rates than expected from the end of February, with 23,946 (23,013 to 24,786) total excess deaths. North-West and North-East regions showed one week lag, with higher mortality from the beginning of March and 6,942 (6,142 to 7,667) and 8,033 (7,061 to 9,044) total excess deaths respectively. We observed marked geographical differences also at municipality level. For males, the city of Bergamo (Lombardia) showed the largest percent excess, 88.9% (81.9% to 95.2%), at the peak of the pandemic. An excess of 84.2% (73.8% to 93.4%) was also estimated at the same time for males in the city of Pesaro (Central Italy), in stark contrast with the rest of the region, which does not show evidence of excess deaths. We provided a fully probabilistic analysis of excess mortality during the COVID-19 pandemic at sub-national level, suggesting a differential direct and indirect effect in space and time. Our model can be used to help policy-makers target measures locally to contain the burden on the health-care system as well as reducing social and economic consequences. Additionally, this framework can be used for real-time mortality surveillance, continuous monitoring of local temporal trends and to flag where and when mortality rates deviate from the expected range, which might suggest a second wave of the pandemic.

**Data Availability Statement:** The raw data are available at https://www.istat.it/en/archivio/240106 (as in line 50 of the manuscript). The pre-processed data, together with ALL the code to

reproduce the results is available on the github repository https://github.com/martablangiardo/ExcessDeathsItaly.

**Funding:** MBI acknowledges partial support from National Institutes of Health, grant number R01HD092580-01A1. Infrastructure support for the Department of Epidemiology and Biostatistics was provided by the NIHR Imperial Biomedical Research Centre (BRC). This work was part supported by the MRC Centre for Environment and Health, which is currently funded by the Medical Research Council (MR/S019669/1). All the authors were supported by core funding from their institutions. The funders had no role in study design, data collection and analysis, decision to publish, or preparation of the manuscript.

**Competing interests:** The authors have declared that no competing interests exist.

# Introduction

The total impact of the COVID-19 pandemic on mortality should be the least controversial outcome to measure. However, its analysis is complicated by the lack of real time cause specific data; a potential additional issue concerns the quality of coding on the death certificates, particularly in the earlier stages of the pandemic. Furthermore, there are important differences in the recording systems, both across and within countries (e.g. in the UK, England and Wales have consistently reported data on daily deaths based on different time of recording in comparison to Scotland and Northern Ireland [1]). In this context, estimating excess deaths for all causes at national level, with respect to past year trends has been used in several countries as an effective way to evaluate the total burden of the COVID-19 pandemic [2–6], including direct COVID-19-related, as well as indirect effects (e.g. people not being able to access healthcare). At the same time, all-cause mortality is not affected by mis-coding on the death certificates [7].

Despite these positive features, this approach can only present global pictures of the total burden of the first wave of the infection. However, to understand the dynamics of the pandemic, we need to analyse data at sub-national level; this would allow to account for geographical differences due to the infectious nature of the disease, as well as those in the population characteristics and health system provision. Additionally, time trends can vary substantially, rendering comparisons even more complex. Recently, health authorities of each country have been urged to release mortality data at high spatial and temporal resolution to evaluate the pattern of the disease and to compare it with the expected trend from previous years [8]. Up to date, to the best of our knowledge, only two papers have analysed mortality at regional level (one in Italy [9] and another in England and Wales [10]), while there are no comprehensive studies that have looked at the impact of COVID-19 at sub-national level.

In this paper, we present the first extensive analysis of excess mortality at sub-national level in Italy, one of the countries with the largest number of deaths for COVID-19 (as of 15 June 2020, 34,345 confirmed fatalities, making it the fourth most affected country in the world, in absolute terms); we consider municipalities as geographical unit and we model weekly number of deaths for all causes. A report by the Italian National Institute of Health [11] showed the characteristics of the epidemics, both in terms of cases and mortality, across Italian regions. Conversely, our objective is to provide the scientific community with a scalable method to obtain high resolution temporal trends of excess deaths across municipalities, in order to highlight similarities and differences in space and time.

# Methods

## Study design

We used a spatio-temporal disease mapping approach [12] to evaluate the excess mortality in Italy at municipality level, while detecting and predicting its evolution on a weekly basis. First, we modelled weekly mortality trends over the years 2016–2019, accounting for air temperature. We used these trends to predict the weekly- and municipality-specific mortality over the period 1 January to 28 April 2020. Then, we compared the observed mortality for this period with the model-based predictions. This allows us to quantify the excess, defined as the number of deaths from all causes relative to what it would have been expected in the absence of the COVID-19 pandemic, based on the model.

We performed separate analyses for males and females, as previous studies have found differences in mortality between sexes [13, 14]. The main results are presented for the total population; we adjust for the age structure at municipality level and across the study period through

an internal standardisation [15]. We calculated age-specific rates (0-14, 15-24, 25-34, 35-44, 45-54, 55-64, 65-74, 75+) across the country for the period under study in 2016–2019 as these represent the all-cause mortality in years without the pandemic. We then applied the age-specific rates to the year-and age-specific population in each municipality to estimate the expected number of cases as denominator.

## Data

We used official weekly mortality data, available from the Italian Institute of Statistics (ISTAT; source: https://www.istat.it/en/archivio/240106) for 2016–2019. While 2015 was also available, we decided to exclude it as it was substantially different from the remaining years due to a much higher winter mortality following low flu vaccination coverage during autumn 2014. The data included counts by age, sex and municipality of residence of the cases. All-cause mortality cases were recorded for the 7,904 municipalities that comprise the whole Italian territory. Administratively, the Italian municipalities are nested into 107 provinces, which are themselves grouped into 20 regions.

We used the weekly deaths in 2020, available from the same source, as a comparison with the values predicted from our model. Currently the 2020 data are only available on 7,251 municipalities, which cover 91.7% of the Italian territory and 93.4% of the Italian population; hence, we limit the comparison to this subset.

To capture the seasonal variability in death counts, we also included air temperature data (at 2 meters height), obtained from Copernicus ERA5 global weather and climate reanalysis dataset [16], which combines a weather model with observational data from satellites and ground monitors to produce global hourly data at a 30 km grid spatial resolution. Operationally, we processed air temperature data within the Google Earth Engine cloud-based platform to obtain weekly series of air temperature for each Italian municipality.

## Statistical methods

**Municipality weekly trends for 2016–2019.** Let $y_{ijtk}$ be the number of deaths in the $i$-th municipality ($i = 1, \ldots, 7904$), nested within the $j$-th province ($j = 1, \ldots, 107$), for the $t$-th week ($t = 1, \ldots, 17$) and $k$-th year ($k = 1, \ldots, 4$, with year 1 corresponding to 2016). In line with the approach typically considered for disease mapping [12], we model the total number of deaths at the municipality level $y_{ijtk}$, separately for males and females, for each week and year using a Poisson distribution with age-adjusted expected number of cases as offset. We specify the following Bayesian hierarchical model on the log mortality relative risk, which is a common approach to overcome the high variability in the estimates driven by the small numbers of cases, due to the high spatio-temporal resolution [17]. Firstly, we model:

$$y_{ijtk} \sim \text{Poisson}(\text{E}_{ijtk}\rho_{ijtk}), \tag{1}$$

where $\text{E}_{ijtk}$ represents the expected number of deaths, while $\rho_{ijtk}$ is the mortality relative risk. To obtain the weekly expected counts $\text{E}_{ijtk}$, we summed across the age groups for each municipality and year and then divide by the number of weeks in each year, as is commonly done in time series analyses.

As for the relative risk $\rho_{ijtk}$, which is the main model parameter, we specify the following log-linear model:

$$\log(\rho_{ijtk}) = \beta_{0k} + u_i + v_i + \omega_{jt} + f(x_{it}) . \tag{2}$$

The year specific intercept $\beta_{0k}$ is defined as $\beta_{0k} = \beta_0 + \varepsilon_k$, where $\beta_0$ is the global intercept, representing the average mortality rate across all municipalities and years, while $\varepsilon_k$ is an unstructured random effect, i.e. $\varepsilon_k \sim \text{Normal}(0, \tau_\varepsilon^{-1})$, representing the deviation from the global intercept. To take into account the spatial correlation between municipalities, we included a Besag-York-Mollié (BYM) [18] specification. This is given by the sum of an unstructured random effect, $v_i \sim \text{Normal}(0, \tau_v^{-1})$ and an intrinsic conditionally autoregressive structure for $u_i$:

$$u_i \mid \boldsymbol{u}_{-i} \sim \text{Normal}\left( \frac{\sum_{j \in D_i} u_j}{n_{D_i}}, \left(n_{D_i} \tau_u\right)^{-1} \right),$$

where $D_i$ is the set of neighbouring areas for the $i$-th municipality and $n_{D_i}$ is its total number of neighbours.

To model the temporal trend, we specify, separately for each province, a weekly random effect $\omega_{jt}$ through a first order random walk (RW1) with precision $\tau_\omega$ common to all the provinces:

$$\omega_{jt} \mid \omega_{j,t-1} \sim \text{Normal}(\omega_{j,t-1}, \tau_\omega^{-1}).$$

Finally, we included in the linear predictor of Eq (2) a non-linear effect $f(\cdot)$ of the average weekly temperature $x_{it}$ of each municipality. In particular, we assume for $x_{it}$ the following second order random walk (RW2) model:

$$x_{it} \mid x_{i,t-1}, x_{i,t-2} \sim \text{Normal}(2x_{i,t-1} + x_{i,t-2}, \tau_x^{-1}).$$

The formulation is completed by specifying the following minimally informative priors: the precisions (i.e. inverse variances) $\tau_\varepsilon$, $\tau_u$, $\tau_v$, $\tau_x$ and $\tau_\omega$ are modelled using a logGamma(1, 0.1), while we assumed a Normal(0, $10^6$) for the fixed effect $\beta_0$. The overall effect of these priors is to essentially stabilise the inference, which is still mainly driven by the observed data, given the large sample size.

**Weekly death prediction for 2020.** Once the model has been fitted to the observed data, we then use Monte Carlo (MC) simulation to obtain $n = 1000$ samples from the (approximate) joint posterior distribution of all the model parameters. We use the MC samples to fully characterise the uncertainty in the estimates and generate samples from the posterior predictive distribution:

$$p(y_{ijt5} \mid \mathcal{D}) = \int p(y_{ijt5} \mid \boldsymbol{\theta}) p(\boldsymbol{\theta} \mid \mathcal{D}) \mathrm{d}\boldsymbol{\theta},$$

where $\boldsymbol{\theta} = (\beta_{0k}, u_i, v_i, \omega_{jt}, f(x_{it}), \tau_\varepsilon, \tau_u, \tau_v, \tau_x, \tau_\omega)$ is the vector of *all* model parameters, whose uncertainty given the observed data $\mathcal{D} = (y_{ijt1}, \ldots, y_{ijt4}, x_{it})$ is averaged out.

All the inferential procedures are carried by using the Integrated Nested Laplace Approximation (INLA) approach [19–21] by means of the R-INLA package www.r-inla.org).

This simulation-based approach to the post-processing of the model output allows to fully characterise the uncertainty in the mortality rates and to propagate it in a principled way through to the predicted mortality counts. The data are freely available from https://www.istat.it/en/archivio/240106. The code is available at https://github.com/martablangiardo/ExcessDeathsItaly.

## Results

We present plots of observed and predicted trends in mortality rates as well as maps and plots of percentage excess mortality, estimated as the difference between the observed number of

deaths and the predicted number under current trends (based on past data and on the assumption that the pandemic did not occur), divided by the observed number.

For males, the total number of deaths in the first four months of 2020 was 122,129, progressing from 6,278 in the week commencing on 1 January to 5,748 in the week commencing on the 22 April; for females the total number of deaths was 129,044, going from 6,633 to 6,614 during the same observation period. Nationally, the highest weekly death toll was of 11,301 in the week of 18 March and of 10,612 in the week of 25 March for males and females, respectively.

We present the trend in mortality rates split by five macro-areas: the North-West (Piemonte, Valle d'Aosta, Liguria); Lombardia; the North-East (Veneto, Trentino-Alto Adige, Friuli Venezia Giulia, Emilia-Romagna); the Centre (Lazio, Marche, Toscana and Umbria); and the South and major Islands (Abruzzo, Basilicata, Calabria, Campania, Molise, Puglia, Sardegna and Sicilia). These are consistent with the Eurostat classification, although we report Lombardia (which would normally be included in the North-West) separately, as it was the most affected region (Figs 1 and 2). For both sexes, the model based on 2016–2019 predicts a slightly decreasing trend for the entire period, with the blue curve and the gray ribbon indicating respectively the mean and 95% interval estimate for the posterior predictive distribution of the number of deaths.

Lombardia is the first region to show a sharp increase from the estimated trend, starting with males in the week of the 26 of February (observed rate = 22.6 per 100,000 vs upper limit of the 95% posterior interval = 22.1 per 100,000, with a posterior probability that the observed rates exceeds the estimated trend equal to 0.996). Mortality rates for males in the North-West and the North-East regions start to deviate substantially from the estimated trend a week after Lombardia (starting 4 March), while the Centre shows a small deviation from the week starting with 11 March and the South does not show substantial differences from the model estimates of the time trend. For females there is a lagged effect of the pandemic, with mortality rates deviating from the expected trend on the week of 4 March in Lombardia and 11 March in the North-East and the North-West. The Centre and South do not show any evidence of excess death. It is worth noting that at the end of our follow up, while sharply falling down, mortality rates are at the level of (or above) those normally observed at the beginning of January.

The extent of geographical differences is clear if looking at the trend in the percent excess mortality by province (Figs 3 and 4 for males and females, respectively). In the maps, darker shades of colour indicate larger percent excess mortality in a given week (we only present the progression from the week commencing on 26 February and for the provinces in the North-West, Lombardia and North-East; maps for all the 107 provinces are presented in S1–S4 Figs). There is a clear increasing darkening of the hot-spots at the heart of Lombardia (starting with Lodi and moving to the provinces of Cremona, Bergamo and Brescia), spreading mostly to the West into Piemonte, South-West into parts of Emilia-Romagna and mainly North-East into Trentino-Alto Adige. Interestingly, despite recording the first COVID-19 death in Italy, the region of Veneto (in the North-East) is affected to a much lower extent, in comparison to neighbouring areas of Lombardia.

Looking at specific municipalities the full extent of the geographical variation is even more striking: for males (Fig 5) the city of Piacenza (in Emilia-Romagna) is the first to reach the peak of the pandemic on the week starting on 11 March, when the observed mortality rates reach 177 per 100,000 residents, corresponding to an excess mortality of 86.9% (77.5% to 94.4%). Bergamo and Pesaro (in the region of Marche, in Central Italy) peak one week later, with mortality rates of 182 and 134 per 100,000, corresponding to an excess mortality of 88.9% (81.9% to 95.2%) and 84.2% (73.8% to 93.4%). Brescia shows a lagged peak, occurring over the

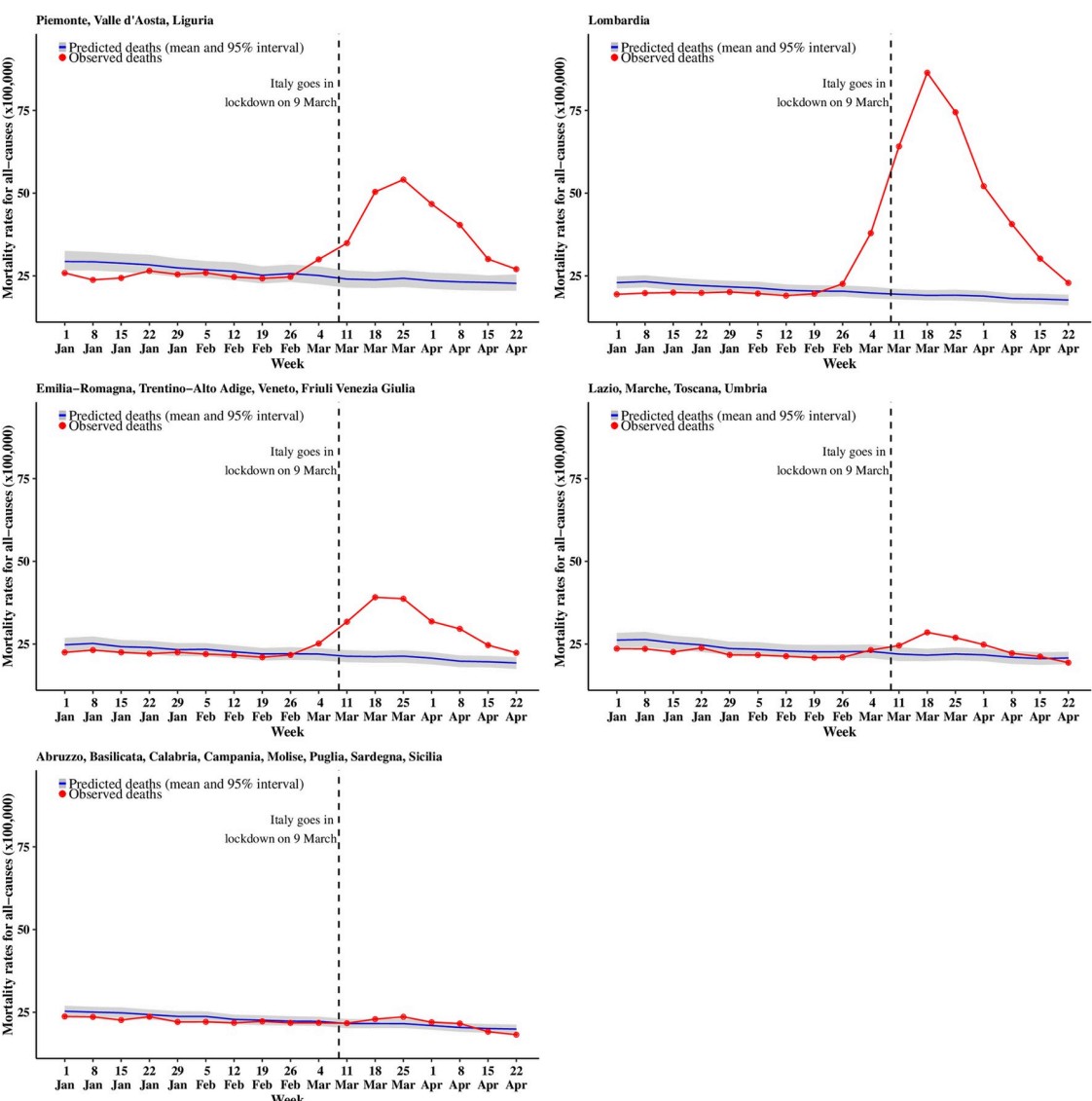

**Fig 1. Trend for all cause mortality (males).** The blue curve shows the posterior mean predicted from the 2016–2019 model, while the gray ribbon describes the posterior 95% interval; the observed number of deaths for 2020 are in red.

weeks of 18 and 25 March and slightly lower mortality rates of 100 per 100,000, with a corresponding excess of 80.6% (70.8% to 88.5%). Meanwhile, Milano, (the second largest city in Italy and the capital of Lombardia) is characterised by a more diluted peak and much lower rates (51 per 100,000) with an excess of 63.2% (55.7% to 70.0%). Finally, Verona (located in Veneto and bordering the Lombard province of Brescia) shows rates barely above the expected interval in the peak weeks. This effect is possibly associated with the comprehensive measures and testing system implemented by the Veneto region to contain the epidemic. [22] Females show slightly lower rates and a peak around the same time, on the week of 18 March. The only exceptions are possibly represented by Milano, which still presents a diluted peak and Verona, which shows no clear evidence of a peak (Fig 6). The plot of the weekly excess mortality for the same municipalities is presented in S5 and S6 Figs.

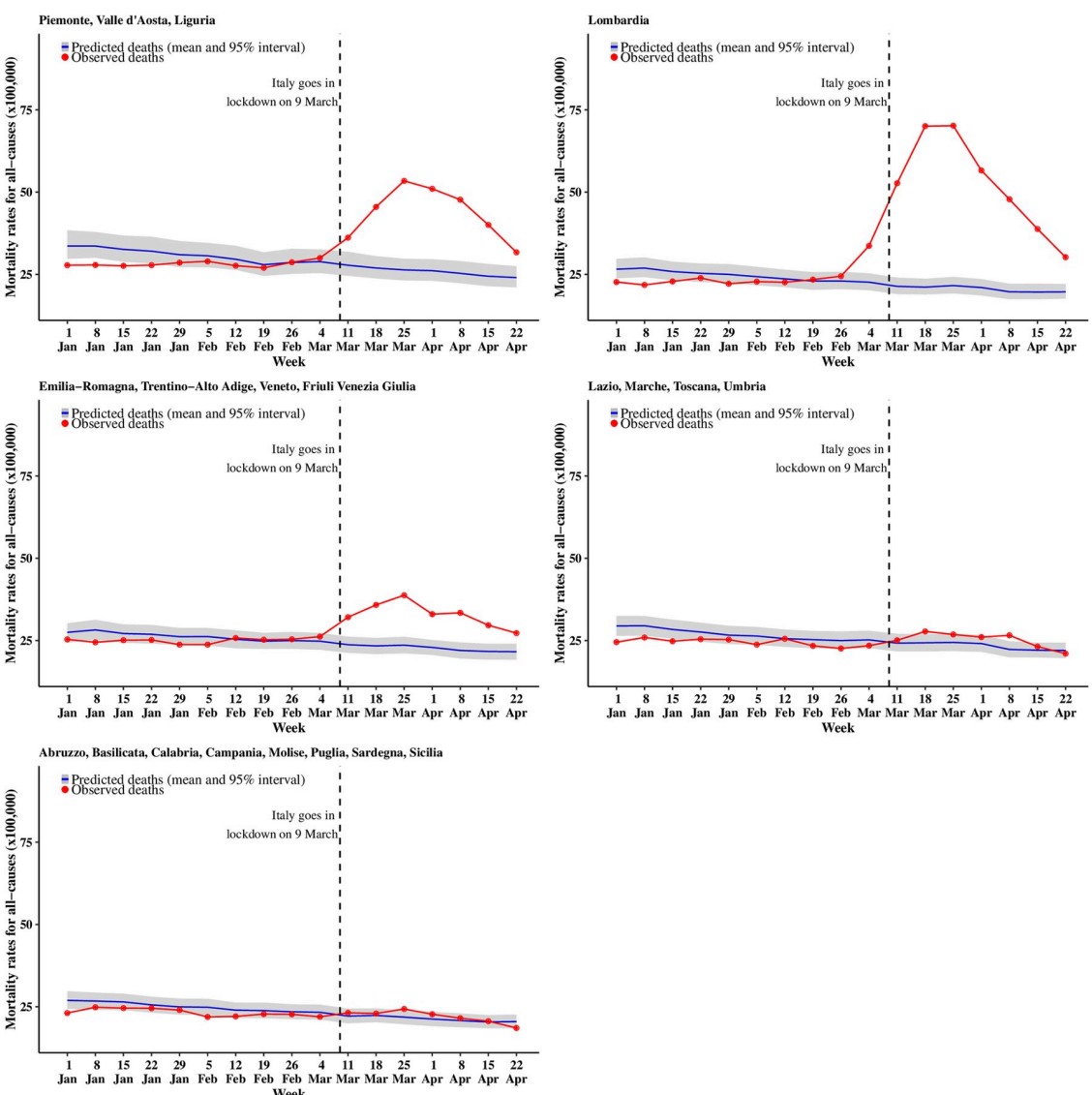

**Fig 2. Trend for all cause mortality (females).** The blue curve shows the posterior mean predicted from the 2016–2019 model, while the gray ribbon describes the posterior 95% interval; the observed number of deaths for 2020 are in red.

We combined the output from our analysis with official data on confirmed deaths for COVID-19 in Italy in the March-April period. Table 1 shows that if we remove the confirmed COVID-19 deaths, Lombardia is still left with a staggering 10,197 (9,264 to 11,037) excess deaths. These may be attributable to the fact that some individuals who would normally access the health care provision were somehow prevented or delayed in doing so during the past few months, resulting in deaths that would have occurred at a lower pace in normal circumstances. Similarly, but to a lower extent, the North-West and North-East show indirect effect of the pandemic on mortality, with an excess, after roughly discounting the confirmed COVID-19 fatality of 2,572 (1,772 to 3,297) and 2,047 (1,075 to 3,058) respectively. The Centre and the South show results that span across 0 in the 95% posterior interval, indicating lack of evidence of any difference in comparison to the estimated trends.

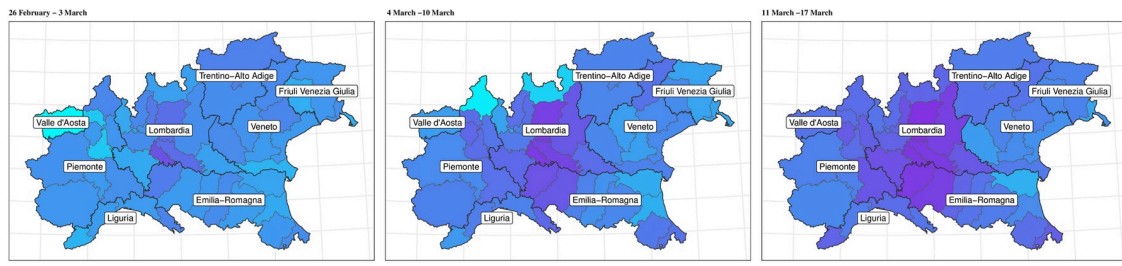

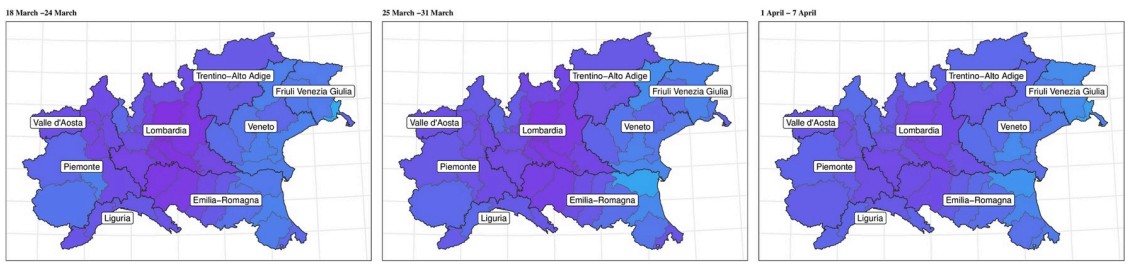

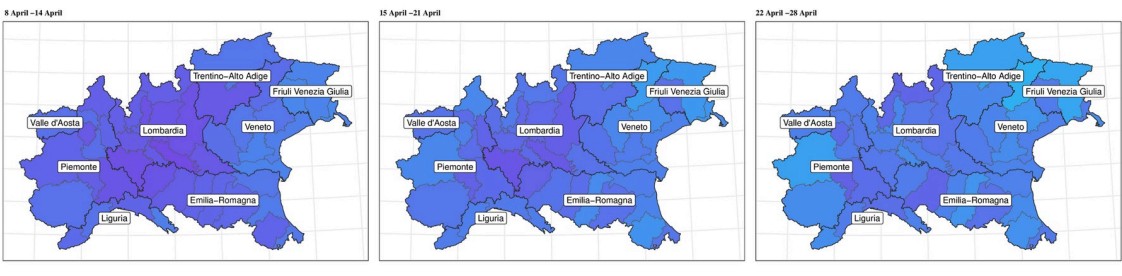

**Fig 3. Percent excess mortality.** Map of the percent excess mortality for the provinces in the areas of North-West, Lombardia and North-East of Italy, weekly posterior predictive mean in 2020 for males. Period: 25 February–28 April.

## Discussion

In this paper we presented the first fully probabilistic analysis of the excess mortality at weekly sub-national level in Italy, one of countries with the highest number of deaths during the COVID-19 pandemic. We found that 2020 starts with lower rates than in previous years, in accordance with [14], but from the beginning of March 2020 there is evidence of an excess mortality compared with the trend in 2016–2019. However, our model suggests large geographical differences in the excess mortality across Italy; while it is now widely established that Lombardia is the most affected region (and one of the hardest hit, globally), we estimated large heterogeneity even with the closest neighbours. For instance, the city of Verona in Veneto, one of the regions in the North East, which borders Brescia (one of the hot-spots in Lombardia)

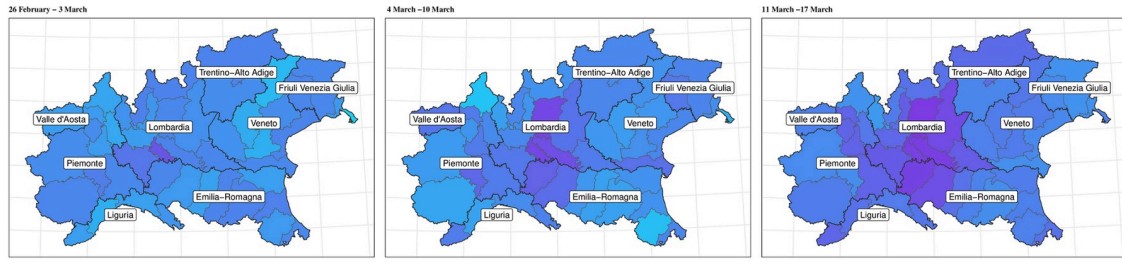

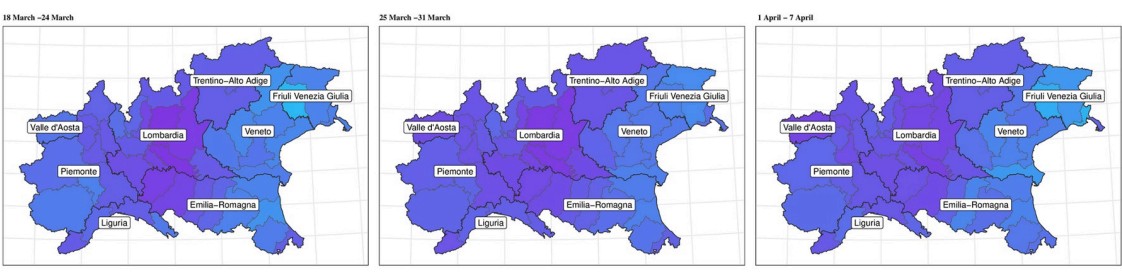

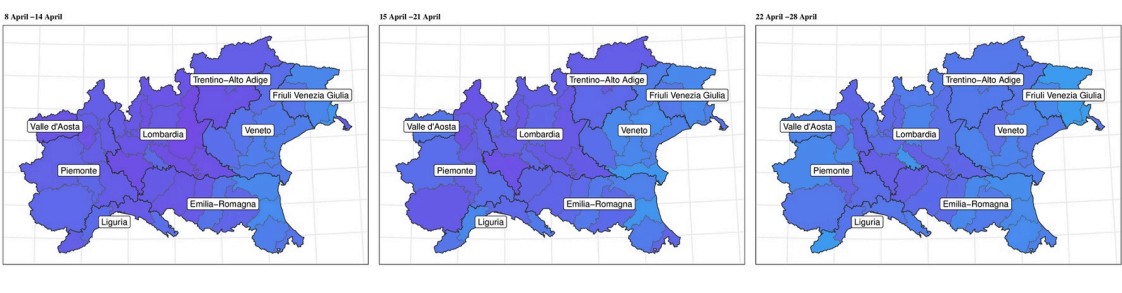

**Fig 4. Percent excess mortality.** Map of the percent excess mortality for the provinces in the areas of North-West, Lombardia and North-East of Italy, weekly posterior predictive mean in 2020 for females. Period: 25 February–28 April.

was associated with mortality rates essentially in line with the historical trend. Similarly, Pesaro (in the central region of Marche) is associated with excess mortality rates that exceed those of Milan (the capital of Lombardia), while belonging to a region that has by and large not been severely affected. We compared our estimated excess mortality with the official statistics reported by ISTAT, which distinguish COVID-19 registered deaths from other causes. While it is impossible, nor is it our objective, to draw causal conclusions from the combination of these data, our analyses suggest a potential large increase in mortality, after discounting the confirmed COVID-19 deaths, particularly in Lombardia, but also in the other regions of Northern Italy. On the other hand, the Centre and the South of Italy do not show evidence of increased net mortality.

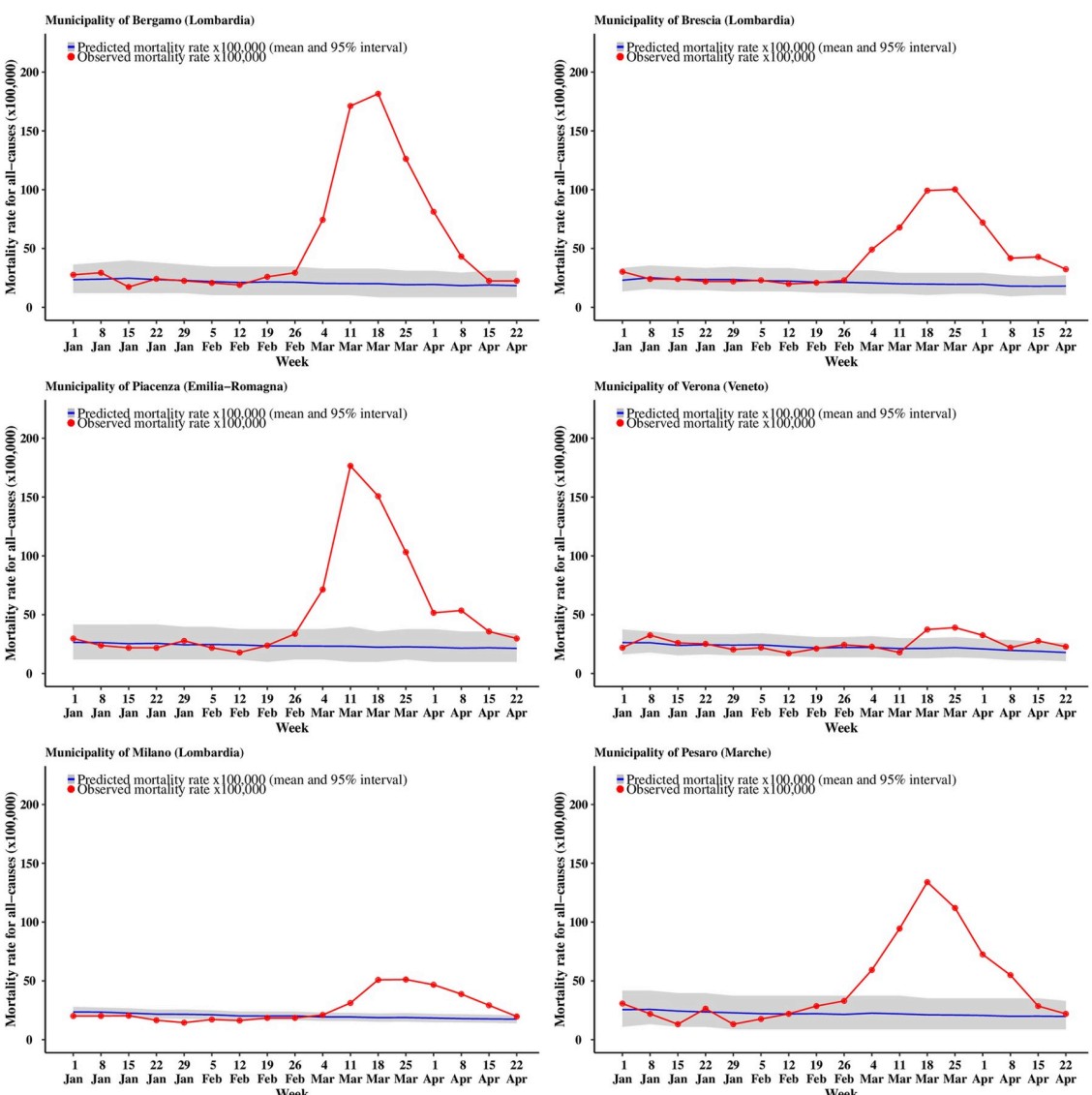

**Fig 5. Trend for all cause mortality (males) in selected municipalities.** The blue curve shows the posterior predictive mean predicted from the 2016–2019 model, while the gray ribbon describes the posterior 95% interval; the observed number of deaths for 2020 are in red.

The specification of the model at municipality level, coupled with the inclusion of weekly non-linear terms, which are allowed to vary for each province, ensures that we can detect heterogeneity across space, while still retaining the flexibility to model the temporal pattern of mortality. This is consistent with typical modelling in disease mapping [23] and surveillance studies [24]. Additionally, we are framed in the same perspective as forecasting studies [25, 26] and we take advantage of the Bayesian nature of our model to characterise the full uncertainty over the estimate of the rates to predict the weekly trends in 2020. Note that, differently from these, our analysis can only be done retrospectively, as estimating excess mortality requires the comparison of the predicted number of deaths with observed ones. At the same time, as our analysis estimates temporal trends for each municipality with the aim of predicting 2020 under the alternative scenario of the absence of a pandemic, we do not need to explicitly

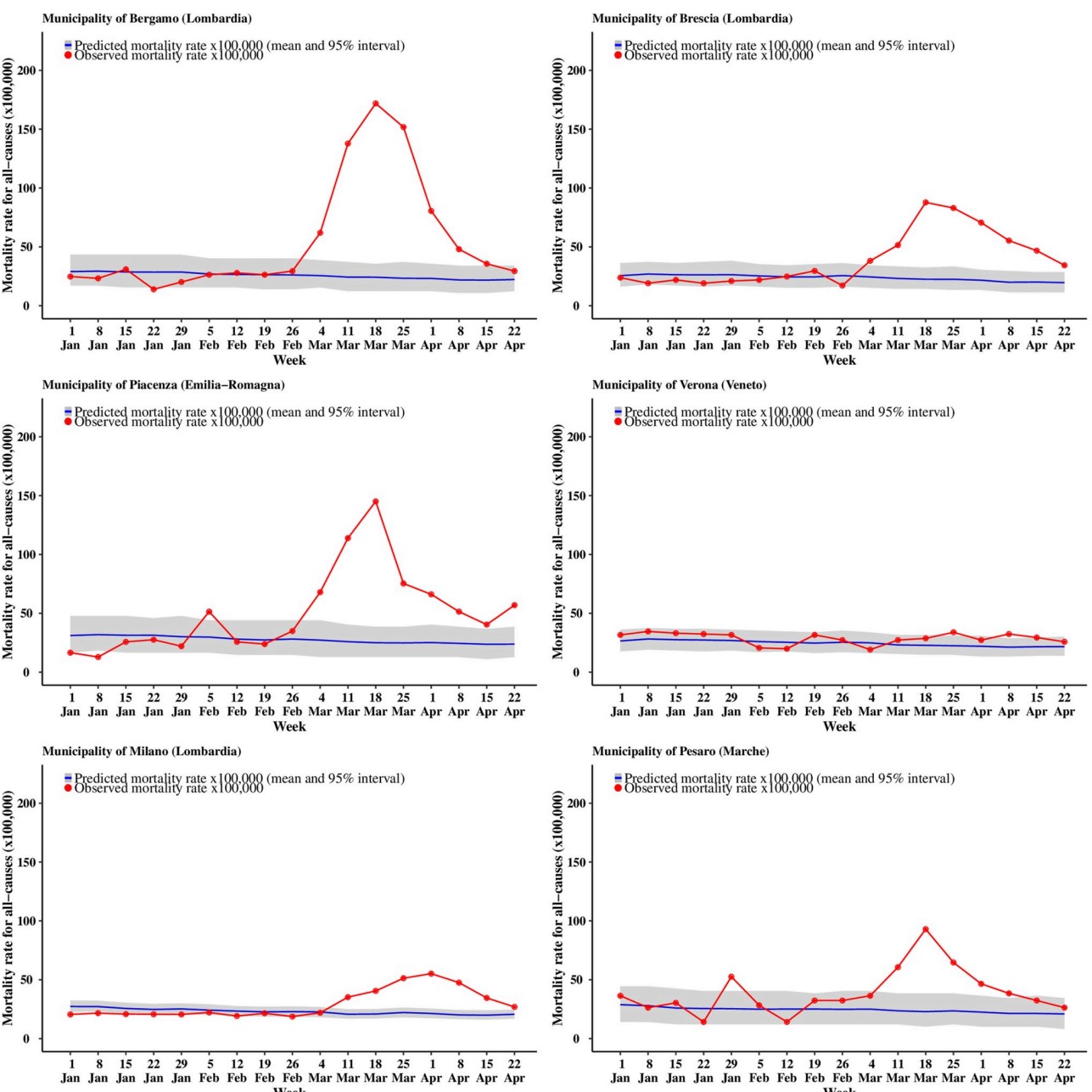

**Fig 6. Trend for all cause mortality (females) in selected municipalities.** The blue curve shows the posterior predictive mean predicted from the 2016–2019 model, while the gray ribbon describes the posterior 95% interval; the observed number of deaths for 2020 are in red.

include covariates only varying in space. This is because their effect is captured by the municipality effect, which estimates the spatial heterogeneity in the rates. Nevertheless, we included weekly mean air temperature for each municipality to adjust the trends in mortality, which is crucial as 2020 was generally warmer than the previous years.

Our model has some limitations. At the time of writing, cause-specific mortality for the year 2020 was not available. Thus, while we were able to estimate the non-COVID-19-related mortality, we cannot disentangle the specific causes behind it. At the same time, we were not able to account for the potential reduction in the number of deaths decreasing during

**Table 1. Observed and modelled excess mortality by macro-regions and by direct COVID-19 fatalities.**

| Macro-region | All cause deaths | Expected deaths posterior mean (95% interval) | Total excess deaths posterior mean (95% interval) | COVID-19 deaths | Non-COVID-19 excess deaths posterior mean (95% interval) |
|---|---|---|---|---|---|
| North West | 18,059 | 11,117 (10,392 to 11,917) | 6,942 (6,142 to 7,667) | 4,370 | 2,572 (1,772 to 3,297) |
| Lombardia | 39,397 | 15,451 (14,611 to 16,384) | 23,946 (23,013 to 24,786) | 13,749 | 10,197 (9,264 to 11,037) |
| North East | 26,694 | 18,661 (17,650 to 19,633) | 8,033 (7,061 to 9,044) | 5,986 | 2,047 (1,075 to 3,058) |
| Centre | 20,903 | 19,315 (18,203 to 20,499) | 1,588 (404 to 2,700) | 2,256 | -668 (-1,852 to 444) |
| South + Islands | 31,367 | 30,846 (29,268 to 32,387) | 521 (-1,020 to 2,098) | 1,577 | -1,056 (-2,597 to 521) |

For the modelled excess we summed the posterior draws across all the municipalities in each macro-area. We present the posterior mean and 95% interval.

lockdown, due for example to a decrease of traffic accidents and injuries at work. This would lead to a conservative estimate of the excess mortality in our model. However, these causes generally account for less than 1% of the total deaths. Our results show that the excess mortality more than offsets that. Additionally, there have been several reports which have showed improvement in air quality during the lockdown in different countries, [27, 28] leading to a consistent reduction of the pollution-related mortality [29, 30]. However, the lack of real-time air pollution data covering the entire study region meant that we could not consider this aspect in the analyses.

While an early comparative study on COVID-19-related mortality suggested that the region Lombardia was not heavily impacted by the epidemic wave [31], in accordance with more recent studies [32, 33], we found that it was the most affected region, for both sexes. Other studies have tried to assess the impact of the pandemic using a lower geographical resolution, e.g. by considering national or regional data [9, 10]. Conversely, our model is specified at the finest administrative level in Italy, which allows us to detect a very large heterogeneity both in space and time, in terms of the spread and the resulting impact of the pandemic on the health-care services. This would not have been picked up by an analysis at regional level.

From the analysis of our results, it appears that the lockdown measures implemented by the Italian government have managed, over the course of several weeks, to contain the excess mortality, as evidenced by the steep decrease during the month of April. However, this is characterised by large spatial heterogeneity: while the Centre and the South return to mortality rates in line with the previous years, the regions of the North-West, North-East and, especially, Lombardia are still showing excess mortality by the end of our observation period. In the most affected region, males mortality rates by the end April return to levels normally observed at the beginning of January, while females rates remain still higher than expected. This suggests perhaps the necessity of staggered relaxation of the lockdown, in order to prevent a second wave and continue to limit the burden on the health-care system. As suggested elsewhere [22], the aggressive programme of mass-testing implemented in Veneto is potentially a driving factor in the much lower excess mortality experienced by the region, in comparison to neighbouring areas such as Lombardia and, albeit to a lower extent, Trentino-Alto Adige.

The first confirmed COVID-19 death in Italy was on 21 of February and from the model there is evidence of a deviation from the expected temporal trend since the last week of February, with the national lockdown implemented on 9 of March. This suggests that our model is able to track the changes in the historical dynamics, thus making our framework particularly valuable to aid policy-making: as new data become available, the model can be used to identify when the rates return to the normal range, but also if and where they start deviating again due to the emergence of a potential second wave requiring for instance social restrictions targeted

in space. Additionally, the modelling framework is a useful tool for prospective real-time surveillance, as it allows to effectively follow each municipality in time and flag unusual behaviours as soon as they happen, aiming at identifying potential new public health threats and contain them.

In addition to providing estimates for the specific Italian case, the proposed modelling framework is highly scalable and adaptable. While we can quantify the geographical differences in excess mortality, understanding the reason why we observe such differences is not trivial. These might be due to the combination of several environmental, social and healthcare related factors; nevertheless, a natural extension of our work consists in the analysis of cause-specific mortality, when the data become available; this will lead to a better understanding of the trends in COVID-19-related mortality, with respect to other competing mortality risks, with a view of subsequently focus on investigating the reasons behind the geographical differences. Moreover, it would be possible to replicate almost without changes the model for other countries, once data become available at a relevant spatial resolution.

## Conclusions

We have used a Bayesian hierarchical modelling framework to describe the evolution of all-cause mortality in Italy from 2016 to 2020 at municipality level and we have pointed out the geographical and temporal differences in the excess mortality during the COVID-19 pandemic. In particular, we have uncovered a striking geographical pattern across Italy, going from the North (severely affected) to the South (which was almost not at all affected). Moreover, we found that the North-West and North-East regions, especially Lombardia, were characterised by a slow return to the expected mortality rate levels, with a persistent excess of mortality by the end of our observation period. Conversely, the Centre and the South did not show evidence of any difference in comparison to the estimated trends.

By highlighting and characterising heterogeneity even between close neighbouring municipalities, our findings suggest the needs of well-targeted responses to COVID-19 pandemic, with flexible and coordinated national and sub-national intervention strategies. They are also particularly valuable for policy makers, when adopting control measures that have the potential of heavily affect the health-care system as well as trigger social and economic consequences. From a methodological point of view, our proposed probabilistic approach represents an effective real-time mortality surveillance tool, which, allowing for a continuous monitoring of localised temporal trends, is able to flag where and when the mortality rates deviate from the expected range, suggesting a successive wave of the pandemic, therefore indicating the necessity of a timely intervention.

## Supporting information

**S1 Fig. Map of the percent excess mortality for the 107 Italian provinces.** Weekly posterior predictive mean in 2020 for males. Period: 5 February–28 April.
(TIF)

**S2 Fig. Map of the percent excess mortality for the 107 Italian provinces.** Weekly posterior predictive mean in 2020 for females. Period: 5 February–28 April.
(TIF)

**S3 Fig. Map of the posterior probability of a positive excess of mortality for the 107 Italian provinces.** Weekly posterior mean in 2020 for males. Period: 5 February–28 April.
(TIF)

**S4 Fig. Map of the posterior probability of a positive excess of mortality for the 107 Italian provinces.** Weekly posterior mean in 2020 for females. Period: 5 February–28 April.
(TIF)

**S5 Fig. Weekly percent excess mortality for all cause mortality (males) in selected municipalities.** Posterior mean (blue) and 95% interval (gray ribbon).
(TIF)

**S6 Fig. Weekly percent excess mortality for all cause mortality (females) in selected municipalities.** Posterior mean (blue) and 95% interval (gray ribbon).
(TIF)

## Author Contributions

**Conceptualization:** Marta Blangiardo, Michela Cameletti, Monica Pirani, Gianluca Baio.

**Data curation:** Gianni Corsetti, Marco Battaglini.

**Formal analysis:** Michela Cameletti, Monica Pirani, Gianluca Baio.

**Methodology:** Marta Blangiardo, Michela Cameletti, Monica Pirani, Gianluca Baio.

**Software:** Marta Blangiardo.

**Writing – original draft:** Marta Blangiardo, Michela Cameletti, Monica Pirani, Gianluca Baio.

**Writing – review & editing:** Gianni Corsetti, Marco Battaglini.

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
