## [Decision Letter · Decision Letter 0]

10 Aug 2020

PONE-D-20-18402

Estimating weekly excess mortality at sub-national level in

Italy during the COVID-19 pandemic

PLOS ONE

Dear Dr. Blangiardo,

Thank you for submitting your manuscript to PLOS ONE. After careful consideration, we feel that it has merit but does not fully meet PLOS ONE’s publication criteria as it currently stands. Therefore, we invite you to submit a revised version of the manuscript that addresses the points raised during the review process.

Please address all issues raised by the reviewer..

We look forward to receiving your revised manuscript.

Kind regards,

Lamberto Manzoli, M.D., M.P.H.

Academic Editor

PLOS ONE

Journal Requirements:

2. Please nesure that all data underlying your figures is also presented in tables.

Reviewers' comments:

Reviewer's Responses to Questions

**Comments to the Author**

1. Is the manuscript technically sound, and do the data support the conclusions?

Reviewer #1: Yes

2. Has the statistical analysis been performed appropriately and rigorously? 

Reviewer #1: Yes

3. Have the authors made all data underlying the findings in their manuscript fully available?

Reviewer #1: Yes

4. Is the manuscript presented in an intelligible fashion and written in standard English?

Reviewer #1: Yes

5. Review Comments to the Author

Reviewer #1: This is an important study investigating spatiotemporal differences in excess mortality at sub-national level during the COVID-19 pandemic in Italy. The Authors provided convincing evidence of the large geographical differences in the excess mortality across Italian regions and municipalities. Moreover the probabilistic models for death predictions used in this manuscript can be a valuable tool for policy-makers in monitoring deviations from the expected range in mortality rates.

The paper is well-written, although there are some points for clarification. Specific comments are below.

- This study wasn’t prospectively registered: this aspect should be mentioned among study’s limitations.

- The Abstract should clarify that the percent excess in mortality of 84.2% (73.8%-93.4%) for the city of Pesaro refers to males.

- The Authors should provide a reference for the spatiotemporal disease mapping approach at page 2 line 39.

- The choice of 2016 as the temporal starting point for time series analyses should be justified.

- Some aspects of the results do not have sufficient comment. The above issue represents the weakest point of the manuscript. More specifically the Authors found that, at municipality level, the cities of Bergamo and Pesaro showed the largest percent excess in mortality. Potential explanations of the observed marked geographical heterogeneity in the pattern of mortality should be provided.

6. PLOS authors have the option to publish the peer review history of their article (what does this mean?). If published, this will include your full peer review and any attached files.

Reviewer #1: No

---

## [Author Response · Author response to Decision Letter 0]

21 Aug 2020

The point-to-point response to the reviewer's comments is provided in the file "Responses_to_reviewers.docx"

---

## [Decision Letter · Decision Letter 1]

16 Sep 2020

PONE-D-20-18402R1

Estimating weekly excess mortality at sub-national level in

Italy during the COVID-19 pandemic

PLOS ONE

Dear Dr. Blangiardo,

Thank you for submitting your manuscript to PLOS ONE. After careful consideration, we feel that it has merit but does not fully meet PLOS ONE’s publication criteria as it currently stands. Therefore, we invite you to submit a revised version of the manuscript that addresses the points raised during the review process.

Please revise the Conclusions. Currently, they are too similar to the abstract, and do not list the main findings of the study. Please use the Conclusion section to briefly summarize what the main findings are and their main practical implication.

We look forward to receiving your revised manuscript.

Kind regards,

Lamberto Manzoli, M.D., M.P.H.

Academic Editor

PLOS ONE

Reviewers' comments:

Reviewer's Responses to Questions

**Comments to the Author**

1. If the authors have adequately addressed your comments raised in a previous round of review and you feel that this manuscript is now acceptable for publication, you may indicate that here to bypass the “Comments to the Author” section, enter your conflict of interest statement in the “Confidential to Editor” section, and submit your "Accept" recommendation.

Reviewer #1: All comments have been addressed

2. Is the manuscript technically sound, and do the data support the conclusions?

Reviewer #1: Yes

3. Has the statistical analysis been performed appropriately and rigorously? 

Reviewer #1: Yes

4. Have the authors made all data underlying the findings in their manuscript fully available?

Reviewer #1: Yes

5. Is the manuscript presented in an intelligible fashion and written in standard English?

Reviewer #1: Yes

6. Review Comments to the Author

Reviewer #1: (No Response)

7. PLOS authors have the option to publish the peer review history of their article (what does this mean?). If published, this will include your full peer review and any attached files.

Reviewer #1: No

---

## [Author Response · Author response to Decision Letter 1]

21 Sep 2020

We have modified the Conclusions as requested (see the tracked changes version).

---

## [Editor Report · Decision Letter 2]

24 Sep 2020

Estimating weekly excess mortality at sub-national level in

Italy during the COVID-19 pandemic

PONE-D-20-18402R2

Dear Dr. Blangiardo,

We’re pleased to inform you that your manuscript has been judged scientifically suitable for publication and will be formally accepted for publication once it meets all outstanding technical requirements.

Kind regards,

Lamberto Manzoli, M.D., M.P.H.

Academic Editor

PLOS ONE
---

## [Editor Report · Acceptance letter]

2 Oct 2020

PONE-D-20-18402R2 

Estimating weekly excess mortality at sub-national level in Italy during the COVID-19 pandemic 

Dear Dr. Blangiardo:

I'm pleased to inform you that your manuscript has been deemed suitable for publication in PLOS ONE. Congratulations! Your manuscript is now with our production department. 

Kind regards, 

on behalf of

Dr. Lamberto Manzoli 

Academic Editor

PLOS ONE